# AE-FLOW: Autoencoders with Normalizing Flows for Medical Images Anomaly Detection

**Yuzhong Zhao, Qiaoqiao Ding, Xiaoqun Zhang**
School of Mathematical Sciences, MOE-LSC and Institute of Natural Sciences
Shanghai Jiao Tong University
`{zhaoyuzhong, dingqiaoqiao, xqzhang}@sjtu.edu.cn`

## Abstract

Anomaly detection from medical images is an important task for clinical screening and diagnosis. In general, a large dataset of normal images are available while only few abnormal images can be collected in clinical practice. By mimicking the diagnosis process of radiologists, we attempt to tackle this problem by learning a tractable distribution of normal images and identify anomalies by differentiating the original image and the reconstructed normal image. More specifically, we propose a normalizing flow-based autoencoder for an efficient and tractable representation of normal medical images. The anomaly score consists of the likelihood originated from the normalizing flow and the reconstruction error of the autoencoder, which allows to identify the abnormality and provide an interpretability at both image and pixel levels. Experimental evaluation on four medical and one non-medical images datasets showed that the proposed model outperformed the other approaches by a large margin, which validated the effectiveness and robustness of the proposed method.

## 1 Introduction

Medical anomaly detection (Taboada-Crispi et al., 2009; Fernando et al., 2021) is an important task in clinical screening and diagnosis by capturing distinctive features in collected biomedical data, such as medical images, electrical biomedical signals or other laboratory results. Anomaly detection aims to detect data that significantly deviates from the majority of data instances, arising in clinical applications due to imbalance between normal and abnormal data and variability of anomaly in real world scenario. Different to usual classification models used for computed aided diagnosis, anomaly detection is usually considered in an unsupervised or semi-supervised paradigm. In this paper, we mainly focus on anomaly detection from medical images, to mimic the diagnosis process of radiologists.

The traditional techniques for finding anomalies are divided into several categories, such as statistics-based methods (Hido et al., 2011; Rousseeuw & Hubert, 2011), distance-based methods (Knorr et al., 2000; Angiulli et al., 2005), density-based methods (Breunig et al., 2000), and clustering-based methods(Yang et al., 2009; Al-Zoubi, 2009), etc. Deep learning for anomaly detection (Wang et al., 2019; Chalapathy & Chawla, 2019; Pang et al., 2021), also known as deep anomaly detection, typically consists of learning a feature representation model of normal images and constructs an anomaly score function for abnormal images by neural networks. There are two main types of approaches for image anomaly detection, one is reconstruction-based model and the other is likelihood-based model in the literature.

The standard procedure of reconstruction-based methods is to first learn an auto-encoder (AE) (Kramer, 1991) or generative models (Goodfellow et al., 2014) for normal images and the difference between the test and reconstructed (generated) images through the representation neural networks can be used to characterize the level of anomaly. For example, AnoGAN (Schlegl et al., 2017) is a generative adversarial networks (GAN) based model utilizing a generator for image reconstruction and an anomaly score using a weighted sum of residual socre and discrimination score. In Akcay et al. (2018), GANomaly considers the distance in the latent feature space to distinguish the anomaly data. F-anoGAN (Schlegl et al., 2019) is an improved version of anoGAN, which si-

multaneously guides encoder training in both image and latent spaces. Even reconstruction-based methods could give the dissimilarity in pixel level, many limitations exists for anomaly detection tasks. The limited capabilities of AEs in modelling high-dimensional data distribution often lead to inaccurate approximations and erroneous reconstructions, such as features are smoothed out in the reconstructed images (Kingma & Welling, 2013; Schlegl et al., 2019; Ravanbakhsh et al., 2017) and image boundaries are predicted as anomaly pixels and appeared in the difference (Schlegl et al., 2017; 2019).

The second type of anomaly detection method is to construct a likelihood function of extracted image features. For example, normalizing flow-based anomaly detection models have been proposed for industry anomaly datasets (Rudolph et al., 2021; Gudovskiy et al., 2022; Yu et al., 2021). Normalizing flow (Dinh et al., 2014; Rezende & Mohamed, 2015; Dinh et al., 2016; Kingma & Dhariwal, 2018) is a popular method that transform the observed data to a tractable distribution. The idea behind normalizing flows is deploying a sequence of invertible and differentiable mappings to transform a complex distribution into a simple probability distribution (e.g., a standard normal distribution). NICE (Dinh et al., 2014) was introduced for modeling complex high-dimensional densities. In Dinh et al. (2016), RealNVP was proposed to improve the coupling layer and a multi-scale architecture was applied to enhance the representative ability of the framework. Recently, Glow proposed in Kingma & Dhariwal (2018) utilizes actnorm and $1 \times 1$ convolution to simulate more realistic images. Normalizing flows have been applied in many fields such as image generation, noise modeling, video generation (Papamakarios et al., 2021). Recently, some anomaly detection methods based on normalizing flows have also emerged and have achieved very high performance for industrial datasets. Differnet (Rudolph et al., 2021) used a backbone network to extract multi-scale features of the input and a normalizing flow network to maximize the likelihood. The anomaly score is defined as the average of negative log-likelihoods. CFLOW-AD (Gudovskiy et al., 2022) added positional embedding layers and utilizes conditional flows to complete the location problems. FastFlow (Yu et al., 2021) proposed a lightweight network and further enhances the accuracy. However, these NF-based anomaly detection methods make decisions by estimating the likelihood of the extracted features, and the structural information of images are missing. From this perspective, an auto-encoder architecture can construct a mapping between the hidden feature space and the data space, and enforce the model to learn structural information of the original data.

In this paper, we propose to construct a loss function and anomaly score function by an auto-encoder with the normalizing flow bottleneck, namely AE-FLOW. This model combines the benefits of normalizing flow methods for computing the anomaly likelihood of extracted features at image level, and the interpretability of reconstruction-based methods at pixel-level. The proposed score function takes both computable probability density in feature features and visual structures consistency in image domain into consideration. The model is in a self-supervised paradigm that uses only normal data, which is more adaptive to real world applications. The experiments on four public medical datasets and one non-medical dataset demonstrated the effectiveness and robustness of the proposed method, with a detail comparison to other existing methods.

## 2 METHODS

We propose a method that integrates the difference on image structures of the original and reconstructed image and the likelihood of image features distribution. The proposed pipeline consists of three important components, i.e., encoder-flow-decoder. Intuitively, normal data will be mapped to the high density area of the standard Gaussian distribution as normalizing flows are trained with normal data. In contrast, those abnormal data will be mapped to the tail of the distribution, making it difficult for the decoder network to effectively reconstruct the original image. The loss function of the three-block pipeline takes into account of the dissimilarity between the reconstructed image and the original image in pixel-level and the likelihood of data distribution in feature-level. The anomaly score is composed of the reconstruction error and the flow likelihood for the inference stage. The pipeline is illustrated in Fig. 1, compared to the usual autoencoder neural network. The detailed illustration of the overall network structure is shown in Fig. 2. In the following, we will explain the three blocks in detail.

***Encoder.*** The model starts with a pretrained encoder network for extracting features of the input image. Each input image will be downsampled four times and the extracted feature is low-dimensional

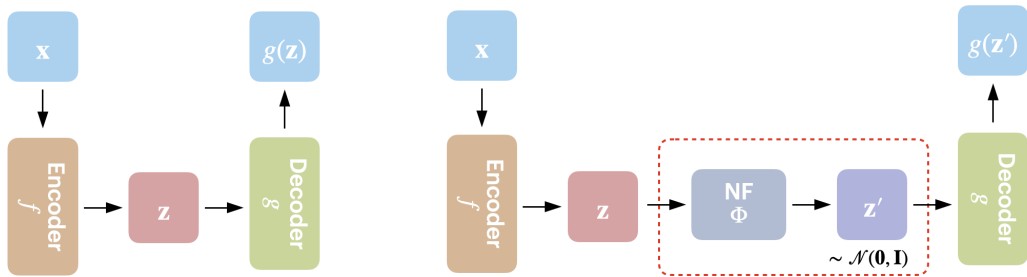

Figure 1: Illustration of the vanilla autoencoder (left) and the skeleton of the proposed AE-FLOW (right). For the vanilla autoencoder, the encoder block extracts the feature of the input and the decoder transforms the intermediate feature back to the image. For the AE-FLOW model, NFs map the extracted feature to the standard Gaussian distribution and the normalized feature is decoded as the reconstructed image.

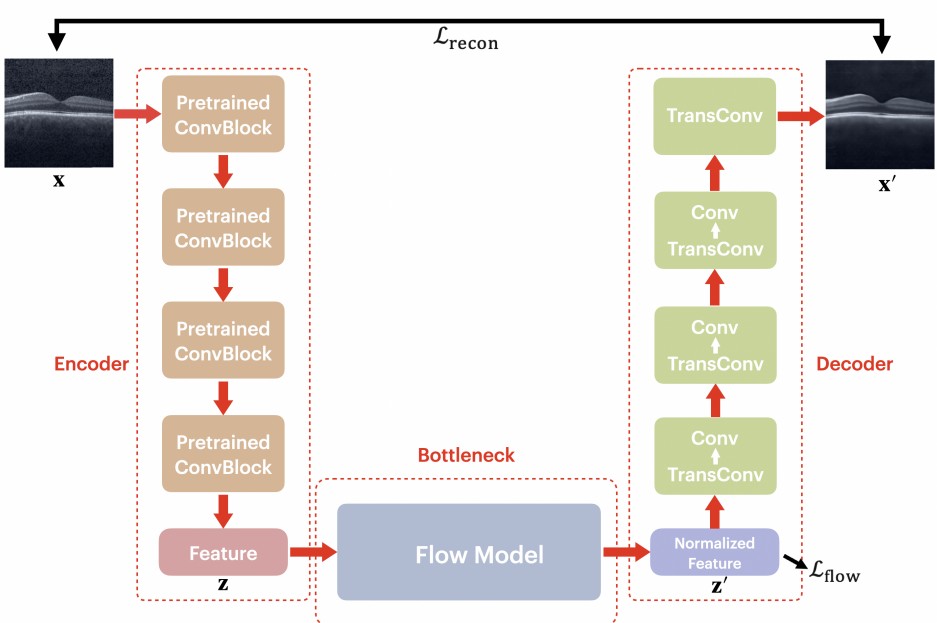

Figure 2: The overview of the AE-FLOW model. Firstly, the encoder block extracts low-dimensional features $\mathbf{z}$ of the input images $\mathbf{x}$. Then, the normalizing flows transform the feature vector to a standard Gaussian distribution. At last, the decoder reconstructs the normalized feature $\mathbf{z}'$ to the image $\mathbf{x}'$. The likelihood of the normalized feature is used as the flow loss and the residual between $\mathbf{x}$ and $\mathbf{x}'$ is used as the reconstruction loss.

and suitable for the flow model training. More specifically, the encoder $f : \mathbb{X} \to \mathbb{Z}$ transforms the input image $\mathbf{x} \in \mathbb{R}^{3 \times H \times W}$ to a feature $\mathbf{z} \in \mathbb{R}^{C \times \frac{H}{16} \times \frac{W}{16}}$, where $H$ and $W$ are the height and width of the original image and $C$ is the number of channels.

***Normalizing flow bottleneck.*** In order to explicitly obtain the probability density $p_{\mathbb{Z}}(\mathbf{z})$ of the extracted feature vector $\mathbf{z}$, we use a normalizing flow model $\Phi : \mathbb{Z} \to \mathbb{Z}'$ to transform it to the feature space $\mathbf{z}' \sim \mathcal{N}(\mathbf{z}'; \mathbf{0}, \boldsymbol{I})$ which follows a standard normal distribution. This flow model is trained by maximizing log-likelihood for the normal image data, which implying a high probability for normal data, while a low probability for abnormal data in the inference stage. The test data can be divided into two categories by applying a threshold on the likelihood. In particular, the normalizing flow map is composed of bijective transformation blocks. For each block, one coupling layer, Softplus activation and output vector permutations are used. The total flow model contains eight flow blocks.

By denoting the input as $\mathbf{x}$ and the output as $\mathbf{y}$ at each flow block, it can be formulated as (Kingma & Dhariwal, 2018; Ardizzone et al., 2018-2022):

$$\mathbf{y} = \boldsymbol{R} \left( \boldsymbol{\Psi} \left( \mathbf{s}_{\text{global}} \right) \odot \text{Coupling} \left( \mathbf{x} \right) + \mathbf{t}_{\text{global}} \right)$$

where $\boldsymbol{R}$ is a $1 \times 1$ invertible convolution that reverses the ordering of the channels, $\boldsymbol{\Psi}(\cdot)$ is the Softplus activation function $\frac{1}{\beta} * \log(1 + \exp(\beta * \cdot))$ with $\beta = 0.5$, $\mathbf{s}_{\text{global}}$ and $\mathbf{t}_{\text{global}}$ are learning parameters corresponding to the scale and bias in the actnorm step, $\odot$ is point-wise product. For the coupling layer, we split the input $\mathbf{x}$ into two compositions $(\mathbf{x}_1, \mathbf{x}_2)^\top$ by the channels, and denote the corresponding output as $\mathbf{u} = (\mathbf{u}_1, \mathbf{u}_2)^\top := \text{Coupling} \left( \mathbf{x} \right)$ and:

$$\mathbf{u}_1 = \mathbf{x}_1 \odot \exp \left( 2 \tanh \left( \mathbf{s} \left( \mathbf{x}_2 \right) \right) \right) + \mathbf{t} \left( \mathbf{x}_2 \right)$$
$$\mathbf{u}_2 = \mathbf{x}_2$$

where $\mathbf{s}\left(\mathbf{x}_2\right)$ and $\mathbf{t}\left(\mathbf{x}_2\right)$ represents the scale and bias which are predicted by one NN, i.e. $\mathbf{s}(\mathbf{x}_2), \mathbf{t}(\mathbf{x}_2) = \text{NN}(\mathbf{x}_2)$. Two subnet architectures are considered for this NN: one follows the network designed by Yu et al. (2021) for which contains two convolutional layers. The second one is based on ResNets (He et al., 2016) structure for a better image structure representation and easier training. Details can be found in the later experiments section.

***Decoder.*** The decoder $g : \mathbb{Z}' \rightarrow \mathbb{X}'$ is used to reconstruct the image $\mathbf{x}'$ from normal distributed data by minimizing the distance between the input image $\mathbf{x}$ and reconstructed image $\mathbf{x}'$. In the decoder, we adopt four modules and each module contains two convolution layers. In each module, we use $3 \times 3$ kernel transposed convolution layer for upsampling operation and another $3 \times 3$ kernel convolution layer to adjust the number of channels and enhance the reconstruction ability.

***Loss Function.*** The loss function consists of two components, the flow loss and the reconstruction loss. The negative logarithm probability likelihood $p_{\mathbb{Z}}(\mathbf{z}')$ is taken as the flow loss. The reconstruction loss is defined as the mean square error (MSE) between the input image and the reconstructed image. We choose MSE in order to force the model to recover main image structures. The weighted sum of two losses will represent the total loss. The loss function with parameter $\alpha$ is formulated as:

$$\mathcal{L} = \alpha * \mathcal{L}_{\text{flow}} + (1 - \alpha) * \mathcal{L}_{\text{recon}}, \tag{1}$$

where

$$\mathcal{L}_{\text{flow}} = -\log p_{\mathbb{Z}}(\mathbf{z}) = -\log p_{\mathbb{Z}'}(\mathbf{z}') - \log \left| \det \frac{\partial \mathbf{z}'}{\partial \mathbf{z}} \right|,$$

$$\mathcal{L}_{\text{recon}} = \text{MSE}(\mathbf{x}, \mathbf{x}') = \frac{1}{N^2} \sum_{i=1}^{N} \sum_{j=1}^{N} (\mathbf{x}_{i,j} - \mathbf{x}'_{i,j})^2.$$

We note that the reconstruction loss can be also replaced with other measures, such as Structural Similarity Index Measure (SSIM) (Wang et al., 2004).

***Anomaly Score.*** Similar to the loss function, we use two components for the anomaly score in the inference step. The negative probability density of the normalized feature is used as the flow score. For the reconstruction loss, we use the negative SSIM between the input image and reconstructed image as its range is of $[-1, 1]$ and it mainly measure the structure difference between images. The window size for calculating SSIM score is set to be 11. The total anomaly score is formulated as a weighted summation of the two scores by a parameter $\beta$

$$\mathcal{S} = \beta * \mathcal{S}_{\text{flow}} + (1 - \beta) * \mathcal{S}_{\text{recon}}, \tag{2}$$

where

$$\mathcal{S}_{\text{flow}} = -p_{\mathbb{Z}'}(\mathbf{z}'),$$
$$\mathcal{S}_{\text{recon}} = -\text{SSIM}(\mathbf{x}, \mathbf{x}').$$

## 3    EXPERIMENTS

### 3.1    DATASETS

The proposed method was performed on five datasets with image-level annotations: the OCT (Optical Coherence Tomography) dataset, the Chest X-ray dataset (Kermany et al., 2018), the skin image

ISIC2018 ( International Skin Imaging Collaboration) dataset (Tschandl et al., 2018; Codella et al., 2019), the brain tumor BraTS2021 (Brain Tumor Segmentation) dataset (Menze et al., 2014; Bakas et al., 2017; Baid et al., 2021) and the microscopic images MIIC (Microscopic Images of Integrated Circuits) dataset (Huang et al., 2021).

**OCT:** The OCT dataset consists of $26,315$ normal images for training. In the test phase, there are four categories: normal, drusen, CNV and DME. The later three categories are regarded as abnormal data and each category has 242 images. The image number of test data for the four categories is chosen to be the same to avoid the data imbalance problem.

**Chest X-ray:** The X-ray chest dataset contains $1,349$ cases of normal images for training. There are 234 normal images and 390 abnormal images which are diagnosed as pneumonia for testing.

**ISIC2018:** The ISIC2018 challenge dataset (task three) contains seven categories and we refer to Lu & Xu (2018) for classifying the NV (nevus) category as normal samples and the rest category as abnormal data. The training set includes 6705 healthy images, and the original validation set are used as the test data, which includes 123 normal images and 70 abnormal images.

**BraTS2021:** The Brain Tumor Segmentation (BraTS) Challenge 2021 dataset consists of 3D multimodal magnetic resonance imaging (MRI) scans. We slice the central portion of each 3D data set to make up the 2D dataset for our model. There are $11,739$ healthy slices in our training set, and $1,051$ slices with tumors and $459$ slices without tumors are used as the test set.

**MIIC:** The MIIC dataset first appeared in (Huang et al., 2021) contains real microscopic images of integrated circuits (ICs), with 23,888 normal images for training, and $1,272$ normal and $116$ abnormal images for testing. We directly compare our methods with the proposed method in the original paper (Huang et al., 2021).

### 3.2 Implementation Details

We note that only normal data are used for the training and only the parameters in the flow model and the decoder are updated for the stability of training process. For the encoder, we chose ImageNet-pretrained Wide ResNet-50-2 (Zagoruyko & Komodakis, 2016) as the feature extractor as it shows to be effective and and provide sufficient reception field. In the following, we provide the network details of the three blocks and the optimizer.

- **Encoder:** The encoder contains four modules, and the feature map of the fourth block were chosen as the extracted features. In more detail, each image is resized to $256 \times 256$ and the size of the extracted feature map is $16 \times 16$ with 1024 channels.

- **Flow:** The flow architecture was constructed using FrEIA library (Ardizzone et al., 2018-2022). Two types of subnets are considered for the coupling layer. One is based on (Yu et al., 2021), for which each block contains two convolutional layers with ReLU activation function, and the corresponding kernel size is $3 \times 3$ and $1 \times 1$ respectively. The other is a ResNet-type network with one $3 \times 3$ convolution layer with batch normalization and ReLU function, and a shortcut connection with $1 \times 1$ convolution will be added as the output. We find the ResNet-type network performs better in most situations, especially when it comes to large-scale datasets. The total flow steps is set to be $8$.

- **Decoder:** The decoder consists of four blocks. We choose transposed convolution with $2 \times 2$ kernel and stride 2 for the upsampling operation, and the channel number is symmetric with the encoder ($1024 \rightarrow 512 \rightarrow 256 \rightarrow 64$). For the former three blocks, we employ another $3 \times 3$ convolutional layer with the same channel number to enhance the ability of reconstructing the image. The final transposed convolution layer with 3 channels will directly output the reconstructed image $\mathbf{x}' \in \mathbb{R}^{3 \times H \times W}$.

- **Optimizer:** The network parameters were acquired via minimizing the loss function using the Adam method (Kingma & Ba, 2014). The momentum, batch size, learning rate and weight decay hyperparameters were set to be 0.9, 64, $2 \cdot 10^{-4}$ and $10^{-5}$ respectively. The last two hyperparameters were set to be $10^{-3}$ and 0 for the Chest X-ray dataset. The model was trained for 100 epochs.

- **Weight parameters:** For the loss function, we find the model attains the best results when weight parameters $\alpha = 0.5$ and $\beta = 0.9$, which we will elaborate more in the later section.

## 3.3 RESULTS

The proposed methods were compared with five deep learning-based approaches: one AE-based method (Kramer, 1991), two representative GAN-based methods: f-AnoGAN (Schlegl et al., 2019) and GANomaly (Akcay et al., 2018): and two recently proposed NF-based methods: DifferNet (Rudolph et al., 2021)and Fastflow (Yu et al., 2021). We refer to Zhao et al. (2021) for the quantitative metrics. All results are reported based on the best performance on the test set. See Tab. 1 for the quantitative comparison of the different methods on the four medical datasets. We adopted Area Under Curves (AUC), f1-score (F1), accuracy (ACC), sensitivity (SEN) and specificity (SPE) as the evaluation metrics and the threshold of ACC, SEN and SPE is determined based on the best F1-score. The threshold used for evaluation is determined based on the best value of F1-score. The corresponding AUC curves for all models are also illustrated in Fig. 3 for a better illustration.

For the OCT dataset, AE-FLOW achieves the best results among all methods in all metrics: AUC, F1, ACC, SEN and SPE. GAN-based methods are superior to AE method and GANomaly outperforms f-AnoGAN. Normalizing flow-based methods, i.e. DifferNet and Fastflow, have shown to be superior while the proposed AE-FLOW method outperforms Fastflow by a noticeably margin with an additional reconstruction block. Thus adding the reconstruction part in AE-FLOW is beneficial to promote the detection accuracy. The Chest X-ray datasets contains significantly less data than the OCT dataset, thus the accuracy are less high than the OCT data for all the methods. We find the performance of GAN-based method can not exceed that of autoencoder on this dataset, which may be due to the insufficiency of training data. However, we find that even with a small dataset, normalizing flow-based techniques may still provide satisfactory results. By taking the advantage of normalizing flows and the reconstruction error measure, our model still achieves the best results in terms of AUC, F1, accuracy and specificity. For the ISIC2018 dataset, the given normal category (nevus) can be easily mistaken with other categories, which makes anomaly detection challenging. For this case, our method still performs significantly superior to other methods, and the impact of $\alpha$ and $\beta$ on the AUC (provided in supplementary material) also demonstrates that the flow score is a robust index. The experimental results of each method for the BraTS2021 dataset are comparatively similar, while our method is still competitive and produces marginally improved results than the others. For the non-medical dataset MIIC, we compare with the method in Huang et al. (2021), and the results are shown in Tab 2. Although the performance of the approach in the literature is impressive, our method is still quite competitive and demonstrates its effectiveness.

Table 1: Quantitative comparison for different methods.

| Dataset | OCT (%) | | | | | Chest X-ray (%) | | | | |
|---|---|---|---|---|---|---|---|---|---|---|
| Metric | AUC | F1 | ACC | SEN | SPE | AUC | F1 | ACC | SEN | SPE |
| Autoencoder | 83.02 | 88.64 | 82.13 | 92.29 | 51.65 | 81.02 | 81.63 | 73.08 | 94.87 | 36.75 |
| f-AnoGAN (Schlegl et al., 2019) | 85.09 | 86.70 | 79.24 | 89.67 | 47.93 | 71.44 | 79.96 | 69.39 | **96.92** | 23.50 |
| GANomaly (Akcay et al., 2018) | 90.52 | 91.09 | 86.16 | 93.53 | 64.05 | 76.14 | 81.18 | 74.36 | 87.69 | 52.14 |
| DifferNet (Rudolph et al., 2021) | 94.27 | 92.85 | 89.05 | 93.80 | 74.79 | 86.41 | 84.62 | 79.17 | 90.77 | 59.83 |
| Fastflow (Yu et al., 2021) | 94.08 | 92.60 | 88.43 | 95.45 | 67.36 | 90.25 | 86.02 | 80.77 | 93.59 | 59.40 |
| Ours | **98.15** | **96.36** | **94.42** | **96.56** | **88.02** | **92.00** | **88.92** | **85.58** | 91.28 | **76.07** |
| Dataset | ISIC2018 (%) | | | | | BraTS2021 (%) | | | | |
| Metric | AUC | F1 | ACC | SEN | SPE | AUC | F1 | ACC | SEN | SPE |
| Autoencoder | 73.97 | 62.57 | 64.77 | 78.57 | 56.91 | 70.38 | 82.47 | 70.79 | **98.38** | 07.63 |
| f-AnoGAN (Schlegl et al., 2019) | 79.80 | 67.03 | 68.39 | 85.71 | 58.54 | 78.06 | 84.89 | 76.42 | 94.77 | 34.42 |
| GANomaly (Akcay et al., 2018) | 72.54 | 60.95 | 56.99 | **90.00** | 38.21 | 74.50 | 82.33 | 70.86 | 97.24 | 10.46 |
| DifferNet (Rudolph et al., 2021) | 76.52 | 64.58 | 64.25 | 87.14 | 51.22 | 78.94 | 83.78 | 74.70 | 93.53 | 31.59 |
| Fastflow (Yu et al., 2021) | 80.81 | 69.95 | 70.98 | **90.00** | 60.16 | 80.94 | 85.37 | 78.21 | 90.96 | **49.02** |
| Ours | **87.79** | **80.56** | **84.97** | 81.43 | **86.99** | **81.58** | **86.06** | **78.74** | 93.91 | 44.01 |

Table 2: Quantitative comparison for MIIC dataset.

| Method | AUC | F1 | TPR | FPR |
|---|---|---|---|---|
| f-anoGAN(Schlegl et al., 2019) | 56.38 | 31.65 | 18.96 | 00.08 |
| GANomaly (Akcay et al., 2018) | 93.34 | 74.64 | 67.24 | 01.18 |
| Proposed in Huang et al. (2021) | 97.28 | 83.48 | 78.45 | 00.86 |
| Ours | **98.69** | **89.62** | **81.03** | **00.07** |

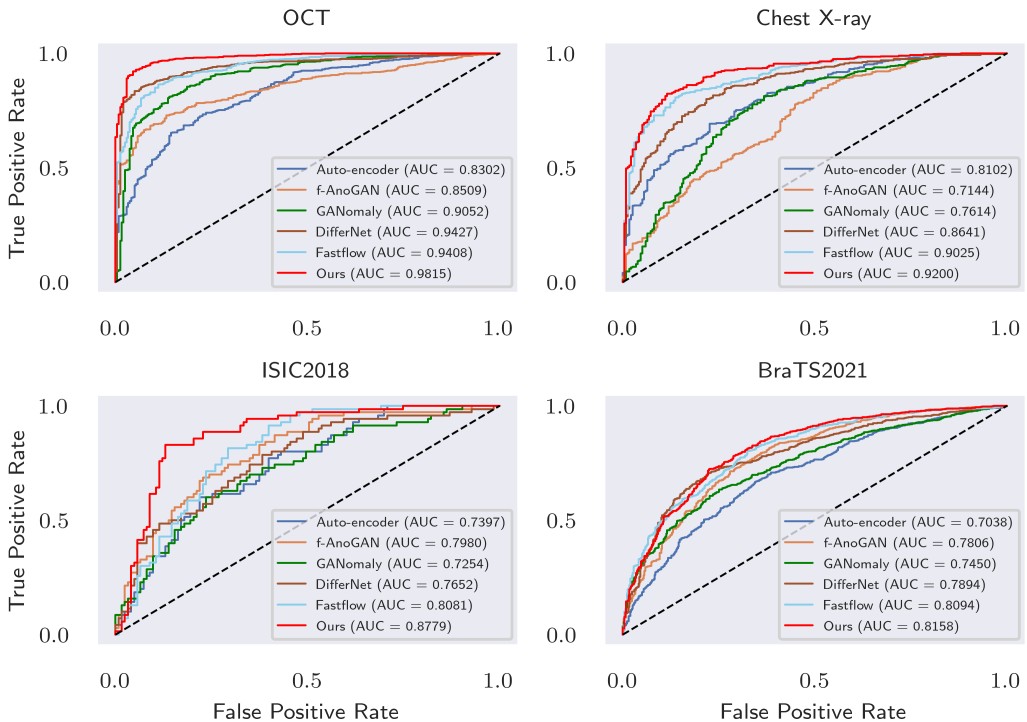

Figure 3: AUC curves for different methods. Our method attains the best AUC score for all four medical datasets.

To further validate the effectiveness, the distribution curves of the computed anomaly score for the two categories, i.e. normal and anomaly are shown in Fig. 4. The horizontal axis of the figure represents the anomaly score and the vertical axis represents the probability density. It is shown that the predicted anomaly score of anomaly data is commonly larger than that of normal data. Moreover, the overlapping area takes a small percentage of the total area, and each curve hits its own unique peak, which demonstrates that the proposed anomaly score function is a valid and effective measure for the classification of normal and abnormal samples.

Compared to flow-based method, the proposed AE-FLOW model allows to provide some visual interpretability of the anomaly detection model. In Fig. 5 and Fig. 6, we show the original, the reconstructed and the residual images for some normal and abnormal samples from the OCT and Chest X-ray datasets respectively. For the OCT dataset, the original images $\mathbf{x}$, reconstructed images $\mathbf{x}'$ and residual images $|\mathbf{x} - \mathbf{x}'|$ are shown in Fig. 5. It shows that the reconstruction image smooth out the zig-zag shape structure, which is identified as the lesion part in the original image. In contrast, the normal data do not contain these structures and the corresponding reconstructed image are much closer to the input images with negligible reconstruction error, and this difference helps to distinguish the anomaly and provide visual interpretability of the detection model. For the chest X-ray dataset, the difference between the original and the reconstructed image for anomaly data is more obvious. Most of X-ray images of pneumonia contain transparent white areas while there are less such structures in normal data. These structures are not present in the reconstructed images based on the AE-FLOW learned from normal data. Thus the differences between the original image and the reconstructed image are in a higher level than that of normal data.

## 3.4 ABLATION STUDY

As the proposed model is composed of the reconstruction error and the flow likelihood for both loss function and anomaly score, we investigated the performance of difference choices of the two weights parameters $\alpha$ and $\beta$ in the loss (equation 1) and anomaly score (equation 2) respectively. Fig 7 gives the AUC value of the proposed model with different $\alpha$ and $\beta$ ranged in $[0, 1]$. When

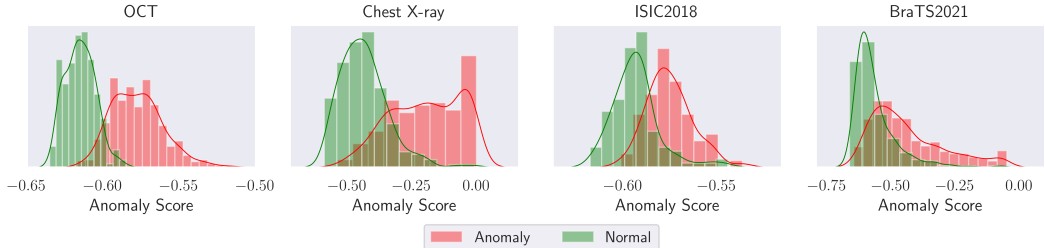

Figure 4: Distribution curves of the anomaly score of the anomaly and normal data. We can see two distributions have different peaks and the ratio of overlapping areas is relatively small, especially in the OCT dataset.

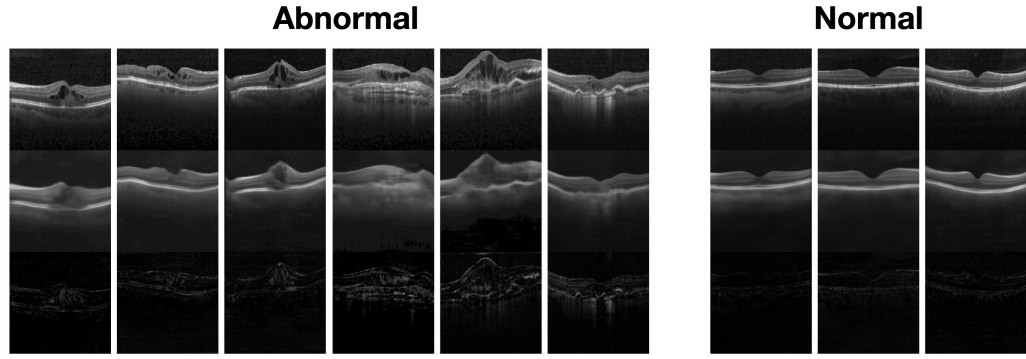

Figure 5: The comparison of normal and abnormal samples from the OCT dataset. For each sample, we show the original, reconstructed and residual images from top to bottom respectively.

$\alpha = \beta = 1.0$, only the flow loss is used for the training and only the flow score is used for anomaly detection. By letting $0.0 < \alpha < 1.0$, the reconstruction error is introduced in the loss function and we can see that the AUC get improved ($96 \rightarrow 97$ for the OCT dataset, $90 \rightarrow 92$ for the Chest X-ray dataset) even if only the flow score is used as the anomaly score ($\beta = 1.0$). This shows that the reconstruction loss ($\mathcal{L}_{\mathrm{recon}}$) positively affects the flow model. Specifically, for both datasets, $\alpha = 0.5, \beta = 0.9$, the best results can be achieved. Furthermore, it can be seen that the proposed model stably attains high AUC for a relatively large range of parameters $\alpha$ and $\beta$, which implies the robustness of the proposed model.

In Tab. 3, we show the AUC score under different settings for the subnet of the NF model and the reconstruction loss function using MSE or SSIM. The results show that the model with these four different combinations still lead to stably high perforamnce, compared to other methods. ResNets with MSE loss achieves the best AUC for the OCT (98.15), ISIC2018 (87.79) and BraTS2021 (81.58) dataset, while 2-Layer CNN with SSIM achievest the best for Chest X-ray (92.00) dataset, which may due to the facts that the anomaly in OCT datasets are mainly due to local lesions while the anomaly in Chest X-ray dataset are more spread out.

Table 3: AUC Comparison under different types of reconstruction loss and NF subnet architecture.

| | OCT (%) | | Chest X-ray (%) | |
|---|---|---|---|---|
| | 2-Layer CNN | ResNets | 2-Layer CNN | ResNets |
| MSE | 96.43 | **98.15** | 91.69 | 89.92 |
| SSIM | 94.89 | 95.31 | **92.00** | 90.47 |
| | ISIC2018 (%) | | BraTS2021 (%) | |
| | 2-Layer CNN | ResNets | 2-Layer CNN | ResNets |
| MSE | 81.19 | **87.79** | 81.55 | **81.58** |
| SSIM | 81.22 | 87.09 | 78.95 | 80.33 |

**Abnormal**                    **Normal**

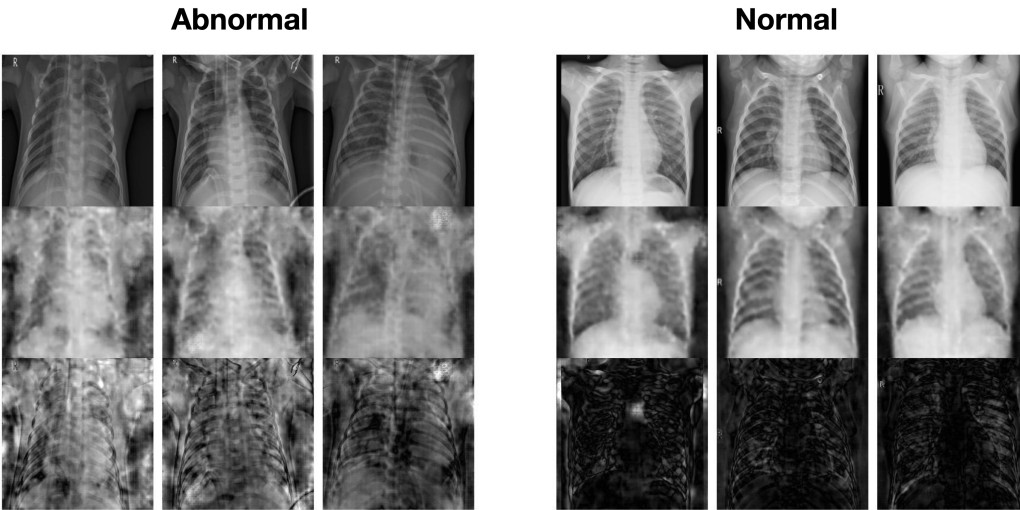

Figure 6: The comparison of the original, reconstructed and residual images (from top to bottom) of normal and abnormal samples from the Chest X-ray dataset.

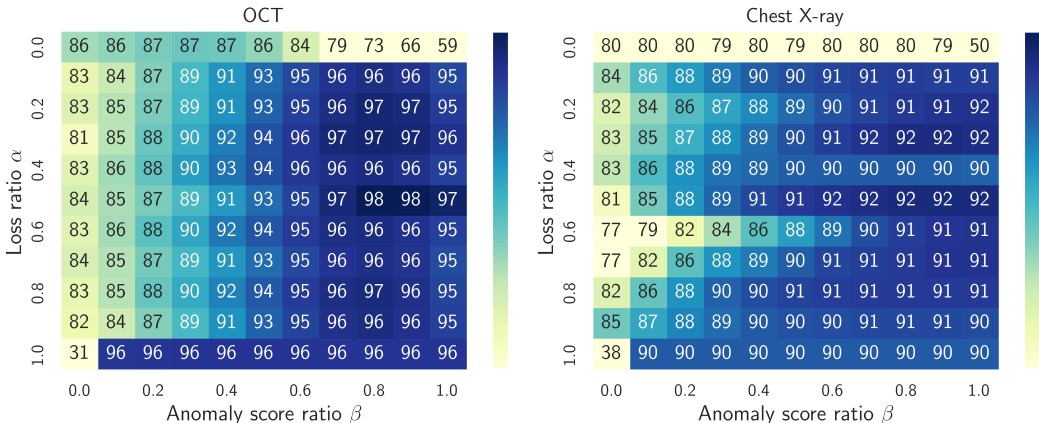

Figure 7: AUC scores under different loss ratio $\alpha$ and score ratio $\beta$ for the OCT (left) and Chest X-ray (right) datasets respectively.

## 4 CONCLUSION

In this paper, we proposed AE-FLOW, a self-supervised model that integrates the tractable likelihood of the NF model and the image dissimilarity between the input and the reconstructed images for the representation of normal data and detecting medical anomalies. The proposed anomaly score function takes into account both the image reconstruction error and the likelihood in feature space for classifying normal and abnormal data, by learning a representation model only from normal data. The method provides not only image-level tractability of normal data but also pixel-level interpretability of anomaly data. Experiments on different medical image datasets demonstrated the effectiveness and robustness of the proposed approach, with a large margin of improvement over the other related and representative approaches for anomaly detection.

ACKNOWLEDGMENTS

This work was supported by Shanghai Municipal Science and Technology Major Project (2021SHZDZX0102) and NSFC (No. 12090024 and 12201402). We thank the Student Innovation Center at Shanghai Jiao Tong University for providing us the computing services.

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

## A   CROSS VALIDATION

We performed five-fold cross-validation experiments for all four medical datasets to further evaluate the robustness and effectiveness of our proposed method. We randomly split the initial test set into validation and test sets in the ratio of 2:8. In order to maintain the balance of each category, we keep the original percentage of normal and abnormal data in each set. The training settings are the same as Section 3.2. We list each metric's means and standard deviations for the five rotations in the Tab. 4.

Table 4: Cross Validation Results for different medical datasets.

| Metric | AUC | F1 | ACC | SEN | SPE |
|---|---|---|---|---|---|
| OCT | $96.78_{\pm0.46}$ | $95.13_{\pm0.38}$ | $92.52_{\pm0.61}$ | $95.56_{\pm1.25}$ | $83.40_{\pm4.58}$ |
| Chest X-ray | $89.77_{\pm1.68}$ | $86.62_{\pm2.35}$ | $81.80_{\pm4.14}$ | $92.18_{\pm2.51}$ | $64.55_{\pm14.20}$ |
| ISIC2018 | $87.37_{\pm3.25}$ | $78.43_{\pm3.20}$ | $81.42_{\pm3.32}$ | $87.86_{\pm1.75}$ | $77.78_{\pm5.11}$ |
| BraTS2021 | $80.22_{\pm0.27}$ | $85.99_{\pm0.45}$ | $78.60_{\pm0.61}$ | $93.89_{\pm1.90}$ | $43.54_{\pm4.53}$ |

As shown in the table, our approach continues to perform well across all datasets. Due to the size of the dataset, the Chest X-ray and ISIC2018 datasets have more fluctuating outcomes than the larger OCT and BraTS2021 datasets.

## B   MORE RESULTS FOR THE RATIOS

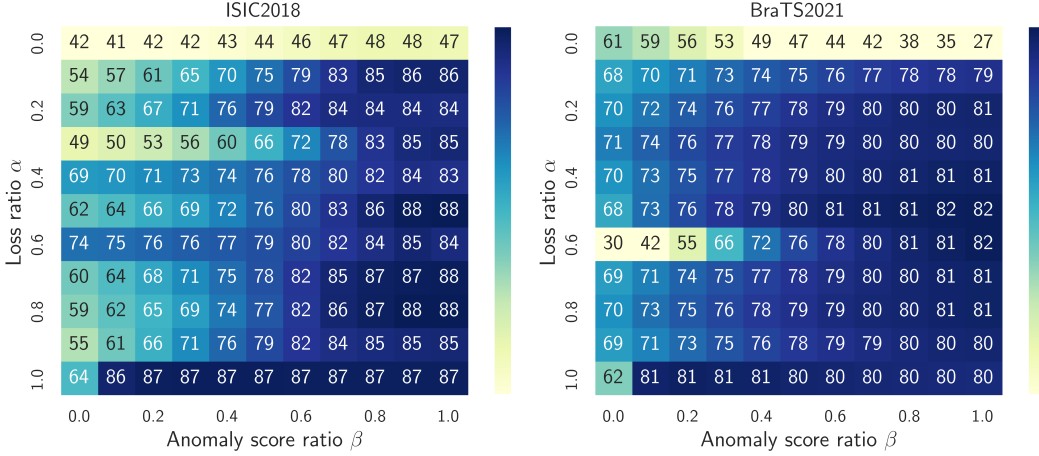

Figure 8: AUC scores under different loss ratio $\alpha$ and score ratio $\beta$ for ISIC2018 and BraTS2021 datasets.

The variations of AUC scores with different loss ratios $\alpha$ and score ratios $\beta$ for the other two medical datasets are displayed in the Fig. 8. Similar to the pattern in Fig. 7, we see that the flow part's accuracy is higher than the reconstruction part's, but after adding the reconstruction part for training ($\alpha < 1$), performance of the likelihood-only anomaly score rises ($87 \to 88$ for ISIC2018 and $80 \to 82$ for BraTS2021). Specifically, we can still achieve best results for $\alpha = 0.5$ and $\beta = 0.9$, and we report our experiments under this setting.

## C   VISUALIZATION FOR MORE DATASETS

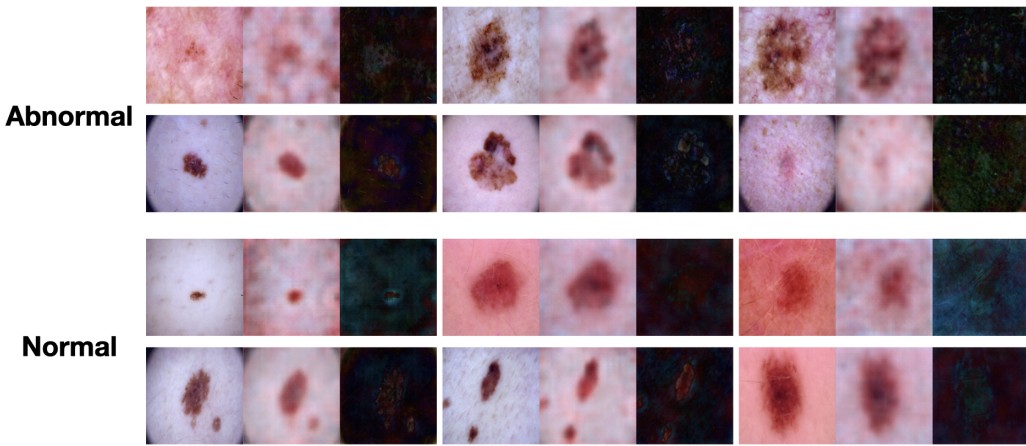

Figure 9: The comparison of normal and abnormal samples from the ISIC2018 dataset. For each sample, we show the original, reconstructed and residual images from left to right respectively.

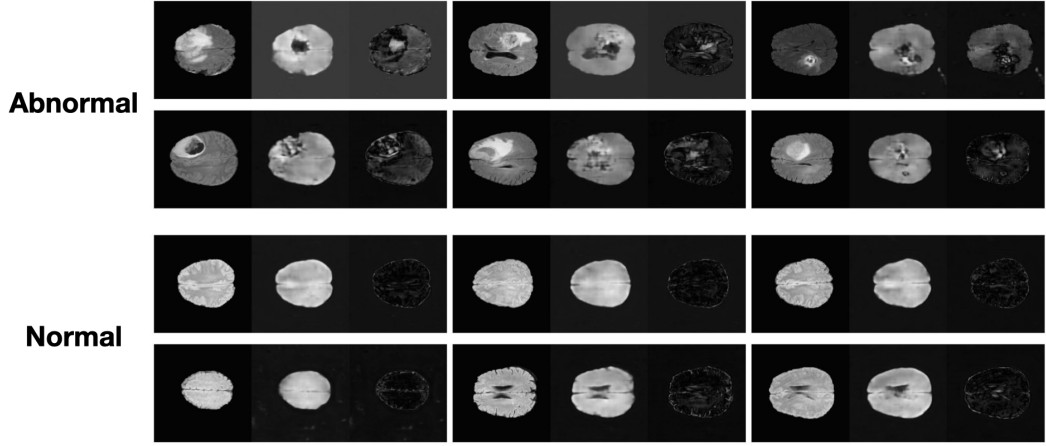

Figure 10: The comparison of normal and abnormal samples from the BraTS2021 dataset. For each sample, we show the original, reconstructed and residual images from left to right respectively.

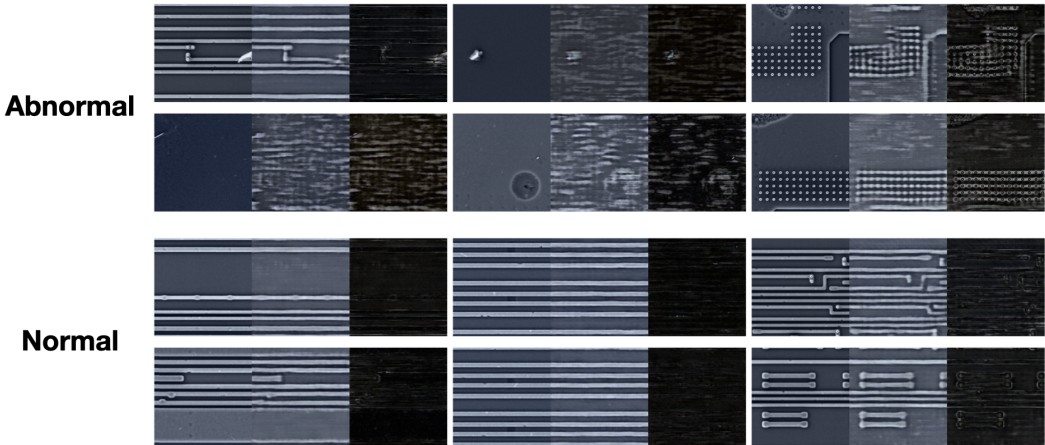

Figure 11: The comparison of normal and abnormal samples from the MIIC dataset. For each sample, we show the original, reconstructed and residual images from left to right respectively.

