# OpenReview forum: "AE-FLOW: Autoencoders with Normalizing Flows  for  Medical Images Anomaly Detection "
_ICLR.cc/2023/Conference — ICLR 2023 poster_

### Official Review · Reviewer_Ngn6 · 2022-10-24

**Confidence:** 3
**Correctness:** 2
**Technical Novelty And Significance:** 2
**Empirical Novelty And Significance:** 3
**Recommendation:** 6

**Clarity, Quality, Novelty And Reproducibility:**

 I think the biggest weakness behind this paper is the lack of novelty. There have been some papers to combine auto-encoder and normalizing flows, and this multi-task training is more common in anomaly detection.

In addition, when a reconstruction loss (such as MSE) is applied to the reconstruction module, overfitting may become a major problem. To my knowledge,  the network could learn an identify mapping more easily.



**Strength And Weaknesses:**

Strengths:

1. This paper adopts a normalizing flow for anomaly detection. The final anomaly score consists of a negative probability density of the normalized feature and reconstruction error, which supervises the model training at both image and pixel levels.
2. Two different datasets (OCT and CT) are used to conduct some comparison experiments to evaluate the effectiveness.

Weaknesses:
1. Direct anomaly detection is little help for medical image diagnosis.  It would be more meaningful to mark the abnormal areas, especially small lesions.
2. It is a standard solution to combine auto-encoder with the normalizing flow for anomaly detection.
3. It'd be better to explain why α and β values are sensitive for the final performance. If we can unify and fix α and β parameters, it will greatly reduce the model training time and will be more available for users.

**Summary Of The Paper:**

This paper proposes a normalizing flow-based autoencoder (AE-Flow) to address medical image anomaly detection.  The AE-Flow model contains three components, an encoder to extract features of the input images, a flow model to convert extracted feature vector to the specific feature space, which follows a standard normal distribution, and a decoder to reconstruct the image from normally distributed data. In the inference stage, a flow score and a reconstruction loss are combined into the final anomaly score. Its contribution to medical image anomaly detection is to adopt the normalizing flow-based method to supervise the model training deeply.

**Summary Of The Review:**

This paper wants to address medical image anomaly detection. The proposed AE-Flow method combines autoencoder and normalizing flows into a multi-task architecture. It is more common for anomaly detection, and could not convince me to accept this paper.

---

> ### Author Response · Authors · 2022-11-18
> **Response to Reviewer Ngn6 (Part1)**
>
> Thanks for your comments.
>
> **Q1**. Direct anomaly detection is little help for medical image diagnosis. It would be more meaningful to mark the abnormal areas, especially small lesions.
>
> **A1**. We strongly disagree with the reviewer on this point. At first anomaly detection is an important step in whole medical diagnosis procedure, for example a fast (and cheap) screening on a large population and rare disease detection before complicated pathological examination.
>
> Regarding marking the abnormal regions, the reality is that not all the diseases present in the form of local lesions or only local lesions delineating does not show any meaningful information of diseases. For example, as shown in the ISIC 2018 skin images, only marking the lesion regions show little information on the difference of abnormal and normal data. For the chest X-ray images, the so-called lesions region is rather global and spread out.
>
> The second application scenario is on abnormality detection based on limited positive samples for the diseases may be rare or the difficulties on collecting a large set of data. An accurate diagnosis based on a supervised method on limited data is not a statically meaningful and trustful method, which cannot be readily used in medical practice.  On the contrary, fast abnormality detection only based on normal data training is more reliable and practical for doctors.
>
> **Q2**. It is a standard solution to combine auto-encoder with the normalizing flow for anomaly detection. I  think the biggest weakness behind this paper is the lack of novelty. There have been some papers to combine auto-encoder and normalizing flows, and this multi-task training is more common in anomaly detection.
>
> **A2**. We don't agree that the proposed framework is a "standard" solution for abnormality detection, especially for medical images data.    We will appreciate if the reviewer can point out any specific related references.
>
> As mentioned above, we found that most industrial anomaly detection methods are based on the likelihood estimation of the extracted features, ignoring the spatial structures of the original image. Medical anomaly detection, on the other hand, often uses implicit generative models such as GANs to reconstruct the image, missing the explicit density estimation of normal data.
>
> The method referenced in [1] is mostly related to our method, but it uses the features extracted by an encoder to reconstruct the input, instead of sampling from the standard normal distribution obtained by the learned normalizing flow. Moreover, the method in [1] does not consider the reconstruction residuals for the anomaly score. While our method was tested on several medical images with real-world applications, the method in [1] was only validated on the experimental dataset MNIST and fashion-MNIST.
>
> Overall, we believe the proposed method is novel by efficiently integrating the global probability learning/sampling and spatial structure preserving, and the extensive numerical results demonstrated the efficiency.
>
> [1] Vanessa M Boehm and Uros Seljak. Probabilistic autoencoder. Transactions on Machine Learning Research, 2022.
>
> **Q3**. Explain why $\alpha$ and $\beta$ values are sensitive for the final performance
> If we can unify and fix $\alpha$ and $\beta$ parameters, it will greatly reduce the model training time and will be more available for users.
>
> **A3**.  We think the reviewer didn't fully understand the table and our statement.  We really appreciate it if the reviewer can take another look. The ablation table on different $\alpha$ and $\beta$ is used to show two points: 1) both the likelihood and reconstruction module are important; 2)  the choice of $\alpha$ and $\beta$ for our model is rather robust as for a large range of $\alpha$ and $\beta$ the accuracy of the method stays stably high. Generally, we find that the model usually produces high accuracy
> with $\alpha$=0.5 and $\beta=0.9$ for several datasets. In practice, we do not need to train for all the parameters. This is further verified on the additional datasets, as shown in Appendix.

---

> ### Author Response · Authors · 2022-11-18
> **Response to Reviewer Ngn6 (Part2)**
>
> **Q4**. Overfitting may become a major problem in network training
>
> **A4**. We disagree on this point and the overfitting problem does not occur in our model.  First of all, the extensive experiments show improved accuracy on test data with and without cross-validation, compared to other SOTA methods. This shows that numerically the method didn't suffer with the over-fitting. Secondly, conceptionally both the normalized flow and the reconstruction network are trained on the normal data, while the unrecoverable structures and out of distribution likelihood of the abnormal data are thus used to effectively identify the abnormality. The results show that abnormality regions in abnormal images are not over-fitted by the proposed reconstruction network and the residual contributes positively to the abnormality score and show the interpretability of the method.
>
> Finally, we will appreciate if the reviewer can give some stronger evidence to support such a **’‘overfitting’‘** declaration on our method.

---

> > ### Comment · Reviewer_Ngn6 · 2022-11-18
> > **Acknowledgement of author's resposes**
> >
> > Thanks for the author's reply. After re-reading the original manuscript and the author's responses,  I have updated my overall score. I agree with your responses and admit its values.

---

### Official Review · Reviewer_hWE4 · 2022-10-24

**Confidence:** 3
**Correctness:** 2
**Technical Novelty And Significance:** 2
**Empirical Novelty And Significance:** 2
**Recommendation:** 6

**Clarity, Quality, Novelty And Reproducibility:**

The paper is well-written and easy to follow, with the exception of a few typos (such as anormaly). The authors make clear and concise statements about their motivation of the study. The authors describe their implementation in detail as well. I believe the study is reproducible. However, the experiments should be conducted on more diverse, large, and public datasets (e.g., skin image dataset, microscopy images, and so on).

**Strength And Weaknesses:**

Strength:
- A promising mechanism focusing on anomaly detection from medical images;
- propose to construct a loss function and anomaly score function;
- a representation model learns only from normal data;
- provides image-level tractability of normal data as well as pixel-level interpretability of anomaly data.

Weaknesses:
- Experiments are conducted in a limited dataset.
- limited technical contribution.


**Summary Of The Paper:**

This paper introduces an auto-encoder with the normalizing flow bottleneck called AE-FLOW for anomaly detection in medical images. The model combines the benefits of normalizing flow methods for computing the anomaly likelihood of extracted features at image level, and the interpretability of reconstruction-based methods at pixel-level.

**Summary Of The Review:**

The study sounds interesting. I think this is a promising mechanism for detecting anomalies in medical images. The problem is that the proposed approach has not been validated with enough experiments and diverse datasets.

It would be nice if the authors could cite a few relevant papers (page 1, second para, first sentence) for their assertions. In my opinion, it is not acceptable by asserting only “One may refer to the review paper (Wang et al., 2019) for more references.” (page1, second para)

We cannot draw conclusions about the robustness of a model based on limited experiments. Thus, I have some reservations on the robustness, quality, and novelty of the study. Overall, I am not fully convinced of the contribution of the paper that would lead me to suggest it for ‘acceptance’.

---

> ### Author Response · Authors · 2022-11-18
> **Response to Reviewer hWE4**
>
> Thanks for your comments and suggestions.
>
> **Q1**. Experiments should be conducted on more diverse, large, and public datasets (e.g., skin image dataset, microscopy images, and so on).
>
> **A1**. As mentioned above, we  added three public  datasets (a skin image dataset, a brain tumor image
> dataset and a microscope dataset), with rather divers characteristic and data sizes. The new experiments further demonstrate the general performance of our methods. Please refer to the updated Tab.1, Tab.2 and Appendix for the results.
>
> **Q2**. It would be nice if the authors could cite a few relevant papers (page 1, second para, fi rst sentence) for their assertions.
>
> **A2**. Thank you for your suggestion. We rephrased some expressions in the paper, and added the missing citations and corrected some typos.
> - Statistics-based methods
>   - Hido et al.2011. Statistical outlier detection using direct density ratio estimation.
>   - Rousseeuw \& Huber. 2011. Robust statistics for outlier detection.
> - Distance-based methods
>   - Knorr et al. 2000. Distance-based outliers: algorithms and applications.
>   - Angiulli et al. 2005. Distance-based detection and prediction of outliers.
> - Density-based methods
>   - Berunig et al. 2000. Lof: identifying density-based local outliers.
> - Clustering-based methods
>   -  Yang et al. 2009. Outlier detection with globally optimal exemplar-based gmm.
>   - Al-Zoubi. 2009. An effective clustering-based approach for outlier detection.

---

> ### Comment · Reviewer_hWE4 · 2022-11-18
> **Acknowledgement on authors' responses**
>
> I really appreciate the authors for taking the time and making the effort to respond to my concerns in this succinct rebuttal phase. It was nice that the authors validated the robustness of their proposed AE-Flow on a variety of relevant datasets. I do not have any further questions at this point. Overall, I am inclined to accept the paper. I have updated my previous score accordingly.
>
> By the way, I can only see the authors' responses for the reviewer "XDoM" in the openreview portal. I was unable to locate the comments for myself (hWE4) or the reviewer "Ngn6." I would like the authors to resolve the issue with the proper selection of "Readers" at their earliest convenience. The authors' responses, though, are visible in my mail. The following are the answers to my concerns:
>
> -----
> Comment title: Response to Reviewer hWE4
>
> Comment: Thanks for your comments and suggestions, and we have revised the manuscript based on your comments as follows.
>
> Q1. Experiments should be conducted on more diverse, large, and public datasets (e.g., skin image dataset, microscopy images, and so on).
>
> A1. We have added three more datasets (a skin image dataset, a brain tumor image dataset and a microscope dataset), with rather diversity characteristic and data sizes. The extensive experiments demonstrate the general performance of our methods. Please see the updated Tab.1, Tab.2 and Appendix for the results.
>
> Q2. It would be nice if the authors could cite a few relevant papers (page 1, second para, fi�rst sentence) for their assertions.
>
> A2. Thank you for your suggestion. We rephrased some expressions in the paper, and added the missing citations and corrected some typos.
>
> *Statistics-based methods*
> - Hido et al.2011. Statistical outlier detection using direct density ratio estimation.
> - Rousseeuw & Huber. 2011. Robust statistics for outlier detection.
>
> *Distance-based methods*
> - Knorr et al. 2000. Distance-based outliers: algorithms and applications.
> - Angiulli et al. 2005. Distance-based detection and prediction of outliers.
>
> *Density-based methods*
> - Berunig et al. 2000. Lof: identifying density-based local outliers.
>
> *Clustering-based method*
> - Yang et al. 2009. Outlier detection with globally optimal exemplar-based gmm.
> - Al-Zoubi. 2009. An effective clustering-based approach for outlier detection.
> -----

---

> > ### Author Response · Authors · 2022-11-18
> > **Response to Reviewer hWE4**
> >
> > We would like to thank the reviewer for the quick response. Indeed, we have updated the general response and the response to the other two reviewers. Now the reader status should be to everyone.  We will appreciate that you check the answers again.

---

### Official Review · Reviewer_XDoM · 2022-10-24

**Confidence:** 4
**Clarity, Quality, Novelty And Reproducibility:** The quality, clarity and originality …
**Correctness:** 3
**Technical Novelty And Significance:** 3
**Empirical Novelty And Significance:** 3
**Recommendation:** 8

**Strength And Weaknesses:**

 Strengths:
1) The paper is very well written, clear, and easy to follow.
2) The figures are well done and informative.
3) The paper addresses a generally important problem for screening of diagnosis images.
4) This is a good use of normalizing flows.

Weakness:
1) I would ask that the authors provide some additional explanation or evidence for the performance boost compared to other benchmarked models. The authors do a good job presenting the results and show good performance over the other baselines; but I am still unsure as why the model is outperforming the others.

2) If one is to use this model for inference on a new example, how would you select a threshold for the anomaly score? The authors seem to have selected an anomaly score based on the testing set F1 scores. For a better model evaluation, identifying an anomaly score using a validation set, then testing on a true held out testing set seems to be the best way to determine the accuracy, sensitivity, specificity, etc. of AE-FLOW. This cross-validation experiment can be added to an appendix.

3) The method applies in general for anomaly detection. Providing results in a non-medical dataset would strengthen the experimental results for AE-FLOW.  This could also be supplemental data.


**Summary Of The Paper:**

The authors present AE-FLOW, an autoencoder-based method for anomaly detection using normalizing flows in the latent space. Anomaly detection is an important problem in medical imaging, becuase in some domains, screening for abnormal diagnoses is critical. The authors propose using normalizing flows to learn an efficient and tractable distribution of medical images. An anomaly score is calculated using both the negative probability density and the negative reconstruction error. The encoder and decoder consist of conventional convolutional layers with downsampling and upsampling, respectively. The normalizing flow bottleneck consists of eight flow blocks. The model contains two major hyperparameters: one that interpolates between the flow and reconstruction losses and one that interpolates between the flow and reconstruction scores. The authors use two medical image dataset, OCT images and chest xrays. They were able to show that AE-FLOW outperforms other anomaly detection baselines on several metrics. An ablation study was performed that that demonstrated both the flow and reconstruction losses are important for overall performance. The authors show qualitative results using residual images generated from the original and reconstruction images. Abnormal images show much large residual values compared to normal images.

**Summary Of The Review:**

The authors present a novel application of normalizing flows for medical image anomaly detection with improved performance over previously benchmarked models.

---

> ### Author Response · Authors · 2022-11-18
> **Response to Reviewer XDoM**
>
> Thanks for your kind comments and constructive suggestions, and we have revised the manuscript based on your comments.
>
> **Q1**. I would ask that the authors provide some additional explanation or evidence for the performance boost compared to other benchmarked models.
>
> **A1**. Due to high computational cost of the Jacobian determinant in normalizing flow, the previous anomaly detection approaches based on the likelihood from the normalizing flow learned from the features extracted by a backbone network without taking into account of the structure of the original images.  With our integrated design, the whole network produces not only global probability distribution based on normalizing flows on the learned features but also local geometrical and spatial structures of normal images, which we believe that capturing both characteristic of normal data are important and improve the effectiveness and robustness of abnormality detection. The numerical results validate the assumptions.
>
>
> **Q2**. If one is to use this model for inference on a new example, how would you select a threshold for the anomaly score? The authors seem to have selected an anomaly score based on the testing set F1 scores. For a better model evaluation, identifying an anomaly score using a validation set, then testing on a true held out testing set seems to be the best way to determine the accuracy, sensitivity, speci city, etc. of AE-FLOW.
>
> **A2**. Thank you for the suggestion. We choose the threshold based on the best value of F1-score previously.  In the revised manuscript, we added 5-fold cross-validation experiments on all four medical datasets (two new ones). The results show that the method still preserves high accuracy, and the relatively small variance shows the stability of the method with respect to different folds of data. Please refer to the section A, Appendix for more details and results.
>
> **Q3**. Providing results in a non-medical dataset would strengthen the experimental results for AE-FLOW.
>
> **A3**.  In addition to two more medical datasets, we further performed the experiments on a non-medical dataset MIIC. For this abnormal detection problem, the method proposed in the original paper performed quite well, but our method can still marginally improve the detection accuracy.

---

### Author Response · Authors · 2022-11-18
**General Response**

We appreciate the kind and constructive comments from the three reviewers. In the following, we first address the **major comments** raised by the reviewers.
- **Test on more datasets**

In order to verify our method on a larger diversity of datasets, we added the experiments on three more datasets, including a skin image dataset ISIC2018, a brain tumor image dataset BraTS2021, and a non-medical microscope dataset MIIC.  For the three additional datasets, the numerical results show that our method still achieved the best results, which demonstrated the general effectiveness and robustness of the proposed method. The quantitative comparisons are present in the updated Tab.1 and Tab.2 and more details
are present in Appendix.

- **Cross-validation**

We performed a five-fold cross-validation experiments on the test data on all four medical datasets. The results show that the method still preserves high accuracy, and the relatively small variance shows the stability of the method with respect to different folds of test data. The details on the mean and the variance on the datasets are present in the section A, Appendix.

- **Technical and empirical novelty**

We agree that our method is conceptional simple, however the method is different from previous methods and empirically shown to be very effective on a diverse set of image data, especially for medical anomaly detection. In the literature, industrial anomaly detection methods  are  usually based on the likelihood estimation of extracted features by a pretrained network, ignoring the spatial image structures of normal image data. Medical anomaly detection, on the other hand, often focus more on implicit generative models such as GANs for image reconstruction, while missing explicit density estimation and sampling of normal data. The novelty of our method relied on proposing an integrated framework by using the power of density estimation and sampling, while reconstructing geometrical consistency of normal image data with decoder from gaussian samples. Numerically, our method was extensively tested and compared on a large set of medical images with real-world applications and a large margin of accuracy improvement were obtained for several datasets. Thus, we believe the proposed method is technically and empirically novel.

In the following, we will address the individual comments from each reviewer.

---

### Decision · Program_Chairs · 2023-01-20

**Decision:**

Accept: poster

**Justification For Why Not Higher Score:**

The novelty of the paper shall be sufficient as a poster, but not that high enough for a higher level as the main components here are not completely new. Instead, it is a combination for a different application.

**Justification For Why Not Lower Score:**

Although normalizing flow is not new, its use for anomaly detection is new. However, this decision can be bumped down if the SAC and PC feel that the contribution is not significant to ICLR.

**Metareview: Summary, Strengths And Weaknesses:**

This paper presents a simple yet effective approach that integrates normalizing flow with auto-encoder for anomaly detection. The strength of the paper is that the paper is well written and easy to follow. The experimental results show obvious improvement.

The main weakness of the paper is the relatively limited novelty as normalizing flow is not new in computer vision. But it is newly used for anomaly detection.

**Note From Pc:**

if the above contains the word "oral" or "spotlight" please see: "oral" presentation means -> notable-top-5% and "spotlight" means -> notable-top-25%. As stated in our emails, we are disassociating presentation type from AC recommendations

**Summary Of Ac-Reviewer Meeting:**

N/A